# Shaping micro-clusters via inverse jamming and topographic close-packing of microbombs

Seunggun Yu[1,2], Hyesung Cho[1,3], Jun Pyo Hong[1], Hyunchul Park[1], Jason Christopher Jolly[3], Hong Suk Kang [3], Jin Hong Lee[1], Junsoo Kim [4], Seung Hwan Lee[1], Albert S. Lee[1], Soon Man Hong[1,5], Cheolmin Park[2], Shu Yang [3] & Chong Min Koo [1,5,6]

Designing topographic clusters is of significant interest, yet it remains challenging as they often lack mobility or deformability. Here we exploit the huge volumetric expansion (up to 3000%) of a new type of building block, thermally expandable microbombs. They consist of a viscoelastic polymeric shell and a volatile gas core, which, within structural confinement, create micro-clusters via inverse jamming and topographical close-packing. Upon heating, microbombs anchored in rigid confinement underwent balloon-like blowing up, allowing for dense clusters via soft interplay between viscoelastic shells. Importantly, the confinement is unyielding against the internal pressure of the microbombs, thereby enabling self-assembled clusters, which can be coupled with topographic inscription to introduce structural hierarchy on the clusters. Our strategy provides densely packed yet ultralight clusters with a variety of complex shapes, cleavages, curvatures, and hierarchy. In turn, these clusters will enrich our ability to explore the assemblies of the ever-increasing range of microparticle systems.

[1] Materials Architecturing Research Center, Korea Institute of Science and Technology, Seoul 136-791, Republic of Korea. [2] Department of Materials Science and Engineering, Yonsei University, Seoul 120-749, Republic of Korea. [3] Department of Materials Science and Engineering, University of Pennsylvania, 3231 Walnut Street, Philadelphia, PA 19104, USA. [4] 3D New Devices Research Section, Electronics and Telecommunications Research Institute, Daejeon 305-700, Republic of Korea. [5] Nanomaterials Science and Engineering, University of Science and Technology, Daejeon 305-350, Republic of Korea. [6] KU-KIST Graduate School of Converging Science and Technology, Korea University, Seoul 02841, Republic of Korea. Seunggun Yu and Hyesung Cho contributed equally to this work. Correspondence and requests for materials should be addressed to S.Y. (email: shuyang@seas.upenn.edu) or to C.M.K. (email: koo@kist.re.kr)

Prescribing nano- and micro-building blocks[1], including spheres[2], rods[3], liquid crystals[4], and biomaterials from proteins[5], viruses[6], and DNA[7], paves the way to program their collective assemblies in the forms of periodic lattices[8], aperiodic quasicrystals[9], "colloidal molecules"[10–12], and supramolecules[13]. One of the prerequisites when guiding the formation of densely packed clusters is that the building blocks should be rigid, incompressible, and impenetrable. The face-centered cubic (FCC) and hexagonal close-packed (HCP) lattices assembled from hard spheres[14] have been exploited for their unique optical[15], electrical[16], and mechanical properties[17]. As such, there has been increasing interest in synthesizing non-spherical particles or clusters for the most efficient space filling but with the least surface area in three-dimensional (3D) Euclidean space, the so-called Kelvin's problem, via soft boundaries[18]. The interplay of facets and corresponding face-selectivity[1], geometry[12], and topology together with long-range soft interactions could provide the basis for, for example, lock-and-key recognition[11, 19] and the formation of more complex phases such as Frank–Kasper phases (or topologically close-packed phases)[20]. Unlike FCC and HCP lattices made of hard spheres, the Frank–Kasper A15 phase, that is, $A_3B$ composition consisting of Kasper polyhedral and icosahedra with larger coordination number[20], can be assembled from soft building blocks, including dendrimers[21], block copolymers[22], and giant tetrahedral polyhedral oligomeric silsesquioxane (POSS) cages[23] with favorable boundary conditions, as a counter example to Kelvin's conjecture.

Meanwhile, jamming[24], a physical process delineated by crowding and kinetic locking of the constituent particles, including granular materials[25], glasses[26], and foams[27], has led to phase transitions where the aggregates behave like a solid[28]. The jamming transition can be controlled by increasing the particle volume fraction $\phi$ within physical confinement by using changes in density $\rho$, temperature $T$, or stress $\sigma$[29], facilitated by rapid impact[30] and the application of vacuum[31] or shear stress[32]. Aiming to bridge the two phenomena, i.e., packing and jamming,

actively and kinetically, we herein introduce thermally expandable microbombs consisting of a polymeric shell and a volatile gas core. The core undergoes a large volumetric change, up to ~ 3000% above the glass transition temperature ($T_g$) of the polymeric shell, in topographical confinement, providing progress toward programming a library of micro-clusters in a 3D space. To exploit the geometric effect of the physical confinement, we design and prepare microwells via soft lithography with a rigid material that could not be deformed by the expansion of the microbombs. In the experiments, our system meets all requirements for jamming[24]: an increase in internal stress ($\sigma \uparrow$), a decrease in inter-particle distance ($d \downarrow$, thus $\phi \uparrow$), and an increase in interaction energy ($U \uparrow$) between components upon heating the microbombs above $T_g$. In the meantime, the softened viscoelastic shell could freely expand until contacting their neighboring microbombs or the walls of the given confinement. As a result, we created a new type of topographic clusters that have the seemingly contradictory characteristics of dense packing ($\phi$ approaching 1) yet an ultralight weight ($\rho \approx 0$, nearly empty, similar to a balloon). For this reason, we refer to the assembly of the microbombs in our system as inverse jamming and topographical close-packing. This strategy leads to a new paradigm of micro-clusters with a wide variety of shapes (e.g., circles, triangles, squares, pentagons, and hexagons in cross-section), partitions (from single to multiple unit cells), edge profiles (from round to sharp corners), and hierarchy.

## Results

**Introduction of microbombs and their expansion behavior.**
Unlike conventional hard spheres, the microbomb-building block used in our system has a thin shell of poly(vinylidene chloride-co-acrylonitrile-co-methyl methacrylate) (PVCAMM) encapsulating isobutane gas that has a low boiling point (b.p.~ − 11.7 °C) as illustrated in Fig. 1a. Upon heating above $T_g$ of the polymeric shell, the microbomb can undergo a huge volumetric expansion (> 3000%,

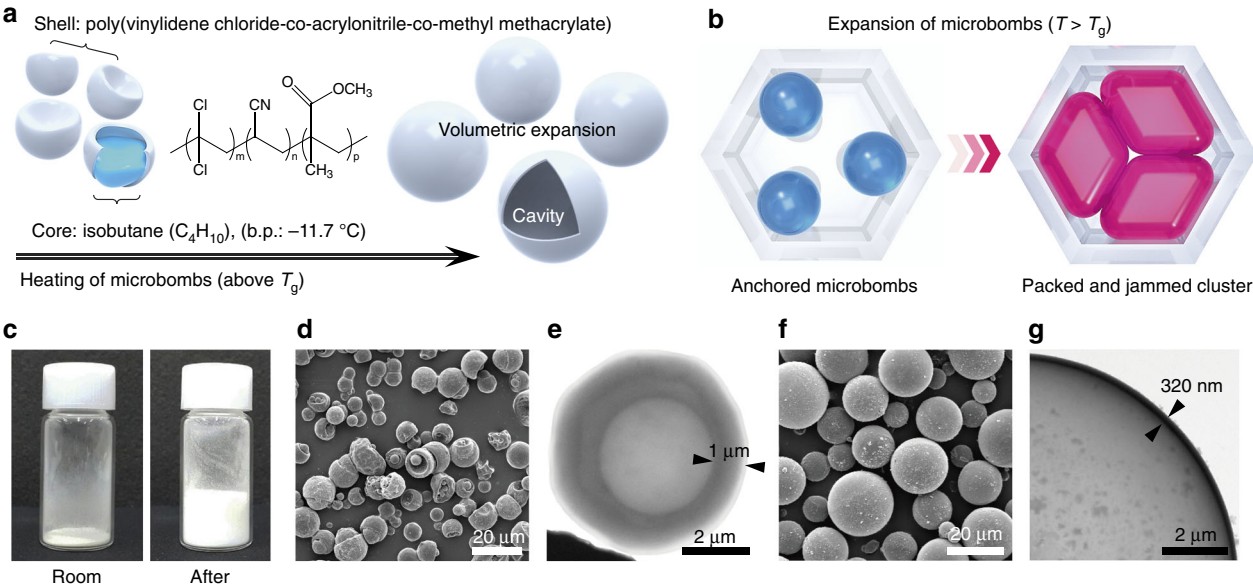

**Fig. 1** Packing and jamming of microbombs within micro-confinement. **a** Illustration of microbombs consisting of an isobutane core and a polymeric shell, which explode and release the isobutane upon heating. **b** Spontaneous packing and jamming of microbombs inside microwells at a temperature above $T_g$. **c** Photos of the microbombs in a vial before and after the volumetric expansion. **d–g** Scanning electron microscopy (SEM) images of the microbombs before (**d**) and after (**f**) explosion, and the corresponding transmission electron microscopy (TEM) images showing the shell thickness before (**e**) and after (**g**) explosion

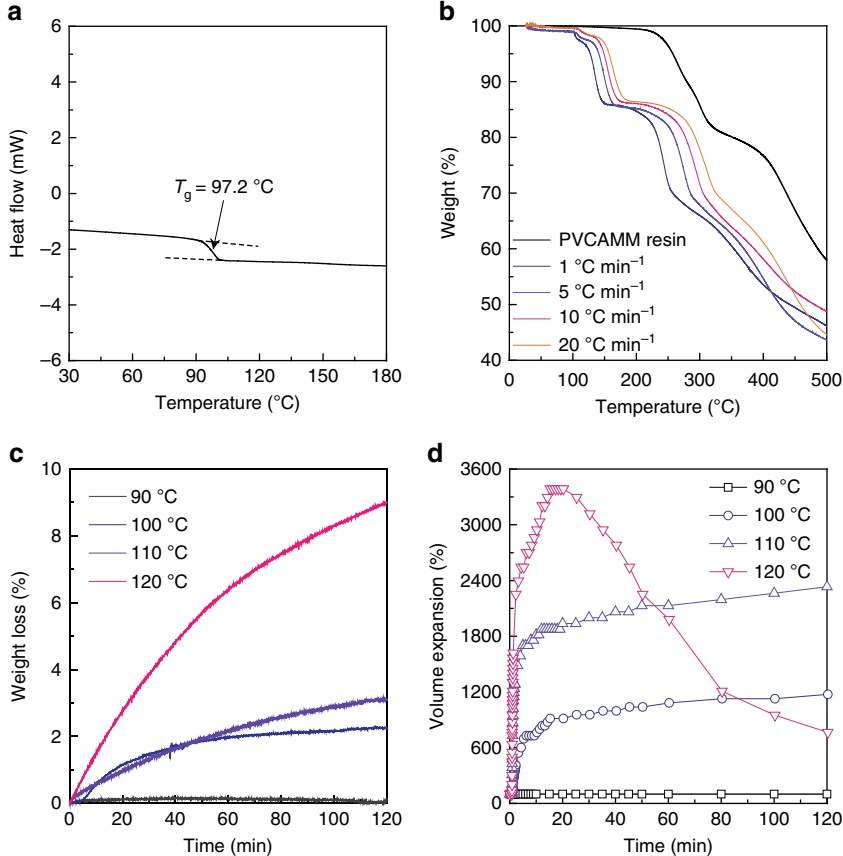

**Fig. 2** Thermal expansion behaviors of the microbombs. **a** Differential scanning calorimetry (DSC) curve of the microbombs, showing the glass transition temperature ($T_g$) ~ 97.2 °C. **b** Thermogravimetric analysis (TGA) curves of the microbombs obtained at different heating rates. **c**, **d** Representative isothermal responses of microbombs at different temperatures: **c** weight loss behavior and **d** volumetric expansion

non-restricted; Supplementary Movie 1) attributed to the volatile isobutane gas expanding the system, much like a balloon. Accordingly, we designed microwells made of rigid polyurethane acrylate (PUA; Young's modulus, $E$ ~ 320 MPa)[33] to confine the internal and kinetic energy generated during the explosion of the microbombs, allowing the topographical assemblies of the micro-clusters to be shaped in a programmed manner as illustrated in Fig. 1b. Restricted by the fixed microwells during the expansion ($T > T_g$), the anchored microbombs have to share the space for optimal space filling, leading to simultaneous close packing and inverse jamming. To confirm the degree of expansion, we first prepared a small quantity of microbombs (~ 10 mg) in a vial at room temperature (Fig. 1c, left) followed by heating them to 140 °C at a rate of ~ 40 °C min⁻¹. As seen in Fig. 1c, right, the overall volume increased ~ 20-fold in the vial. We then examined the topography and size change of the microbombs using scanning electron microscopy (SEM) and transmission electron microscopy (TEM), respectively. The as-received samples had a broad particle size distribution (1–15 µm in diameter; Supplementary Fig. 1), and some were initially crumpled (Fig. 1d). The PVCAMM shell is estimated to be ~ 1 µm thick according to the TEM analysis (Fig. 1e). After heating to 140 °C, all the microbombs were spherical (Fig. 1f), confirming isotropic expansion as a result of the high internal vapor pressure of the isobutane core[34], which exceeded 3584 kPa (Supplementary Table 1). Accordingly, the microbombs expanded up to 35 µm in diameter, while the shell thickness decreased to ~ 320 nm (Fig. 1g).

**Thermomechanical properties of the microbombs.** Understanding the thermomechanical properties of the microbombs is

critical to controlling their expansion, jamming, and packing. A differential scanning calorimetry (DSC) curve showed that the microbomb had a $T_g$ ~ 97.2 °C (Fig. 2a). Thermogravimetric analysis (TGA) curves (Fig. 2b) showed that the microbombs lost weight in approximately three stages. At stage one, up to ~ 100 °C, there was little loss of weight, probably only dehydration of the microbombs. At stage two, up to ~ 220 °C, a steep loss of ~ 15 wt% was observed regardless of the heating rates (1, 5, 10, and 20 °C min⁻¹). The onset temperature of stage two is consistent with the $T_g$ of the polymeric shell. In contrast, the PVCAMM resin (as a control) showed no weight loss up to 220 °C (see the black curve in Fig. 2b). This result indicates that weight loss at stage two is mainly caused by the isobutane gas escaping through the softened viscoelastic shell. Further heating would lead to gradual thermal decomposition of PVCAMM shell (stage three, >220 °C; Supplementary Fig. 2). To further investigate the viscoelastic behaviors of the polymeric shells and the corresponding weight loss, we performed TGA at isothermal conditions under constant temperatures ($T$ = 90, 100, 110, and 120 °C; Fig. 2c). Below $T_g$, the shell remained glassy, which could resist high vapor pressure generated by the thermal expansion of isobutane (~ 1700 kPa; Supplementary Table 1). Therefore, no weight loss (Fig. 2c) or volume change (Fig. 2d) was observed at 90 °C. Above $T_g$, however, the shell softened and became viscoelastic such that it could no longer hold its initial form against the accumulated high internal pressures (2000–2700 kPa), leading to volumetric expansion by 600–3000% (Fig. 2d and Supplementary Fig. 3) in < 5 min. At the highest isothermal temperature ($T$ = 120 °C) in the tests, the size of the microbombs decreased after reaching the greatest expansion due to the leakage of

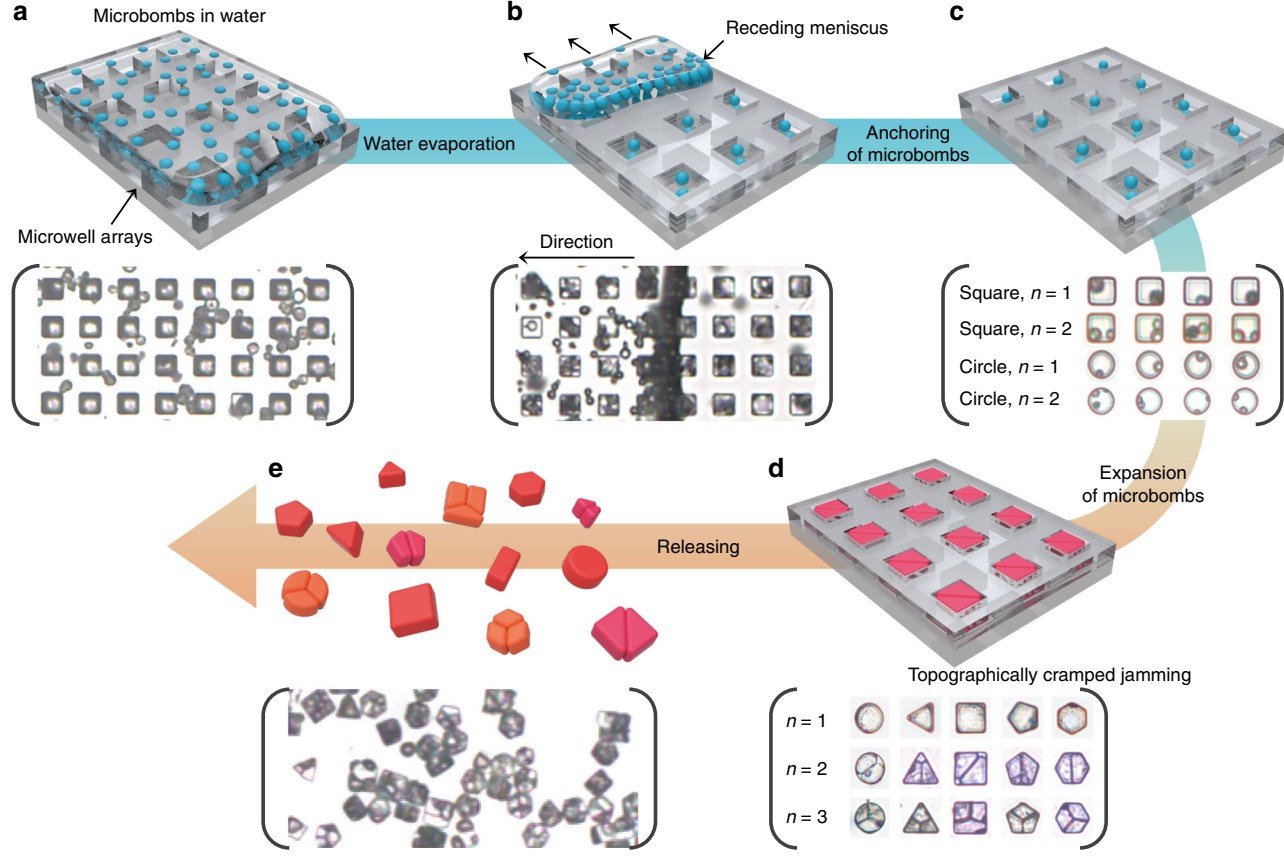

**Fig. 3** Experimental procedures for the topographic confinement of microbombs. The corresponding optical images are presented underneath.
**a** Introduction of the microbomb solution on top of the predefined microwells. **b** Water evaporation and receding meniscus processes to guide the microbomb assembly within the microwells. **c** Anchored microbombs within the microwell array. Optical images illustrate various possible packing combinations of the number of seeds and the shapes of wells. **d** Formation of micro-clusters as a result of dense packing and monolithic jamming of the expanded microbombs in confinement. **e** Release of the clusters utilizing a piece of dissolvable tape

isobutane through the viscoelastic shell (>4 wt%), which in turn would have adverse effects on the jamming and packing; burst clusters were observed (Supplementary Fig. 3d). Therefore, we set the optimal temperature for our studies between 100 and 110 °C, at which the microbombs showed negligible loss of the gas core even after isothermal heating for 2 h.

**Overall procedure for fabricating topographic clusters**. Figure 3 depicts how microbombs could take advantage of both jamming and packing to shape topographic clusters. First, receding meniscus-induced docking[35] was performed to anchor microbombs to the predefined microwell array. Since the pristine samples showed a broad size distribution, we carried out microfluidic sorting[36] to select the seeds with diameter ranging from 4 to 8 µm (Supplementary Fig. 1). Then, the aq. microbomb solution was dispensed onto PUA patterns with various shapes prepared by soft lithography (Supplementary Figs. 4 and 5). Upon water evaporation, the meniscus receded, and the seeds filled the microwells driven by capillary action (Fig. 3b). After waiting for a few tens of minutes for the microbombs to settle (Fig. 3c), a flat polydimethylsiloxane (PDMS) film was placed with conformal contact on top of PUA microwells to seal the wells. Upon heating, the microbombs expanded to fill the rigid wells, which did not change in size or shape during the process (Fig. 3d). In principle, two aspects were presumed to guide the topographic clustering. First, the initial space or vacancy should be less than the total volume of the microbombs in free expansion. Thus, the microbombs could limit one another's expansion, converting

the internal vapor pressure into the stress need to shape the microbomb clusters within the topographic confinement. Second, the expansion should be sufficiently slow to allow the expanded microbombs to detect the neighboring vacancy, via, for example, rolling and lateral movement. For clarification, we refer to the microbombs before expansion as seeds. On the basis of the high mobility of the microbombs during heating, inverse jamming could occur, leading to monolithic integration of the microbombs via viscoelastic shell-to-shell adhesion. As the end of the procedure, the assembled clusters were detached by using adhesive tape and were released by dissolving the tape in water (Supplementary Movie 2). As seen in Fig. 3e, the partitions within the micro-clusters are reminiscent of the expanded seeds.

**Real-time tracking of the deformation of microbombs**. To track the packing and jamming processes, we used SEM to examine the clusters after expansion at different temperatures (90, 100, 110, and 120 °C) within square microwells as shown in Fig. 4a. We observed the seeds but no change in the size or shape of individual seeds at 90 °C, topographic clusters with round and sharp edges at 100 and 110 °C, respectively, and burst seeds at 120 °C. These results are in good agreement with the thermomechanical characteristics of the microbombs presented earlier, while the clusters formed at 100 and 110 °C offer a clue for programming the shape of the assembled clusters topographically. The real-time tracking as shown in Fig. 4b–e revealed the high mobility of the microbombs and the trajectory of their jamming and packing processes. For monitoring, pairs of microbombs were seeded into

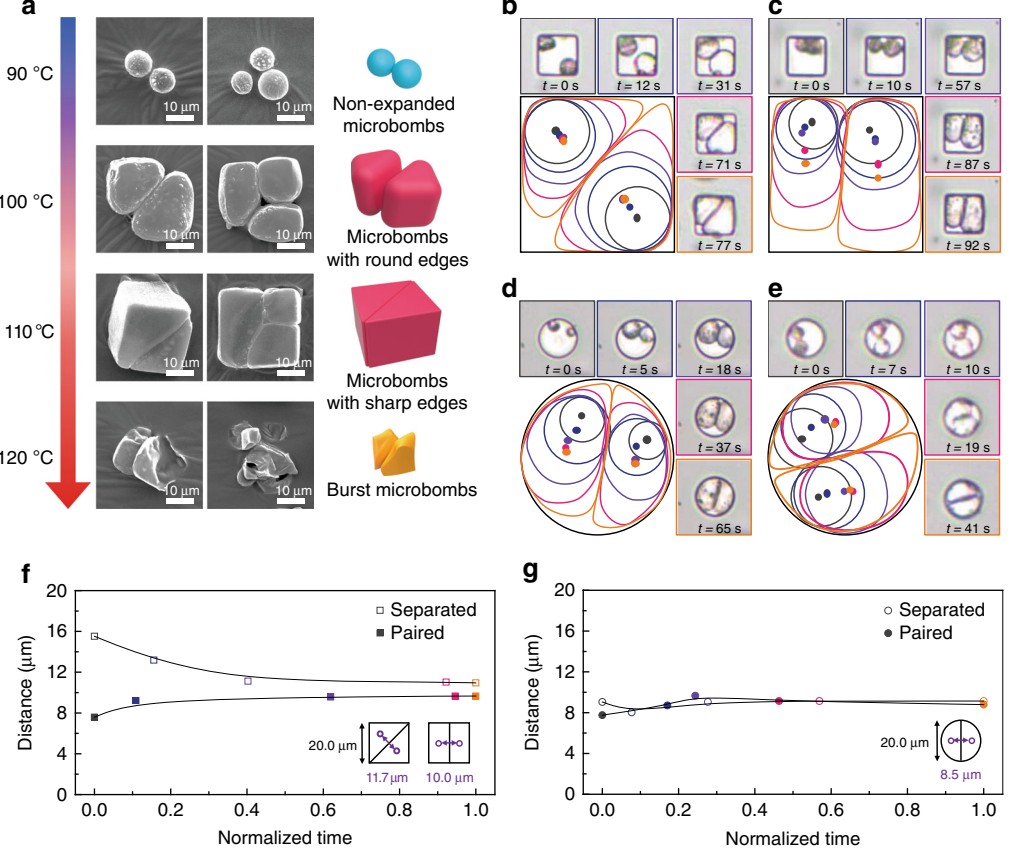

**Fig. 4** Real-time tracking of the micro-cluster formation. **a** SEM images and the corresponding illustrations of the clusters formed at different temperatures. **b**, **c** Optical images and corresponding illustrations of the trajectories of a pair of microbombs that are separated (**b**) and paired (**c**) upon explosion in square confinement. **d**, **e** Optical images and the corresponding illustrations of circular confinement of two separated (**d**) and paired (**e**) microbombs. **f**, **g** Plots of center-to-center distances between two constituent cells as a function of time normalized by the time required to reach the convergent center-to-center distance within square (**f**) and circular (**g**) confinement

two types of wells, square (Fig. 4b, c) and circular (Fig. 4d, e). Within each well, while the total volume fraction $\phi$ of the cluster increased with the expansion of the individual seeds (i.e., inverse jamming), the low surface energy of PUA ($\sim 20\,\mathrm{mJ\,m^{-2}}$)[37] facilitated the microbombs sliding and rolling to make inter-seed contact. It should be noted that once in contact, the viscoelastic shells formed pressure-sensitive shell-to-shell adhesion; the contact area maintained while the internal stress kept the boundary moving toward any neighboring vacancy. Hence, the paired seeds could show limited mobility yet still be active to the direction that has more free space. Soon, severe stress is applied to the surface in contact as the packing of deformable seeds is much dominant than jamming once they recognize the unyielding walls, topographically deforming the shapes of seeds to yield dense packing similar to the foams in expansion[27]. In the square wells, two distinctive final clusters were observed depending on the initial conditions of the seeds. When they were non-contacted (separated) and positioned at opposite vertices (Fig. 4b), discotic expansion occurred until they met each other ($t < 31$ s; Fig. 4b). At $t \sim 77$ s, a diagonal partition between two triangular unit cells (i.e., the expanded microbombs) was generated. When the two seeds were in contact initially (paired), they expanded isotopically in two dimensions (discotic) as they are free even the contact maintained ($t \sim 10$ s, Fig. 4c). The cluster then asymmetrically expanded side by side by deforming the shells to fulfill the dense-pacing in the well, forming two rectangular cells ($t \sim 92$ s). The evidence of jamming is much clear in addressing initial interaction between the two separated seeds as

seen in Fig. 4d (see the black and red circles, $< 5$ s, and Supplementary Fig. 6). We attribute such distinctive differentiation in clustering to the pinning of the seeds at the vertices. In contrast, half-and-half partition of the microbombs within a cluster was always observed in the circular well regardless of the initial positions of the seeds (Fig. 4d, e). Theoretically, in square confinement, the center-to-center distance between two triangular cells is 11.7 µm (defined as $\left[2 - \sqrt{2}\right]L$, where $L$ is the length of the side of the square, 20 µm). In the case of two rectangular cells, the distance is 10 µm. As seen in Fig. 4f, the experimental results are in good agreement with the theoretical prediction. Interestingly, the two curves obtained from the cells in circular confinement converged at the early stage of the process, $< 0.4$ (a time normalized to the on-set time when the center-to-center distance starts to converge) vs. 0.6 in square confinement, implying that the microbombs were much more mobile in the circular well, where only one degree of freedom is programmed in the topographic clustering process. We elucidate further design rules to define degrees of freedom of clusters in the next section.

**Programming the degrees of freedom in topographic clusters.** Empirically, the final clusters can be programmed by the number of seeds ($N = 1, 2, 3, \ldots$) and the number of vertices of the well ($V = 0, 3, 4, 5,$ and 6 for circles, triangles, squares, pentagons, and hexagons, respectively) providing topographic confinement. As shown in Fig. 5a, we analyzed over 2000 clusters. In the confinement of $N = 1$, single-cell particles could be created by

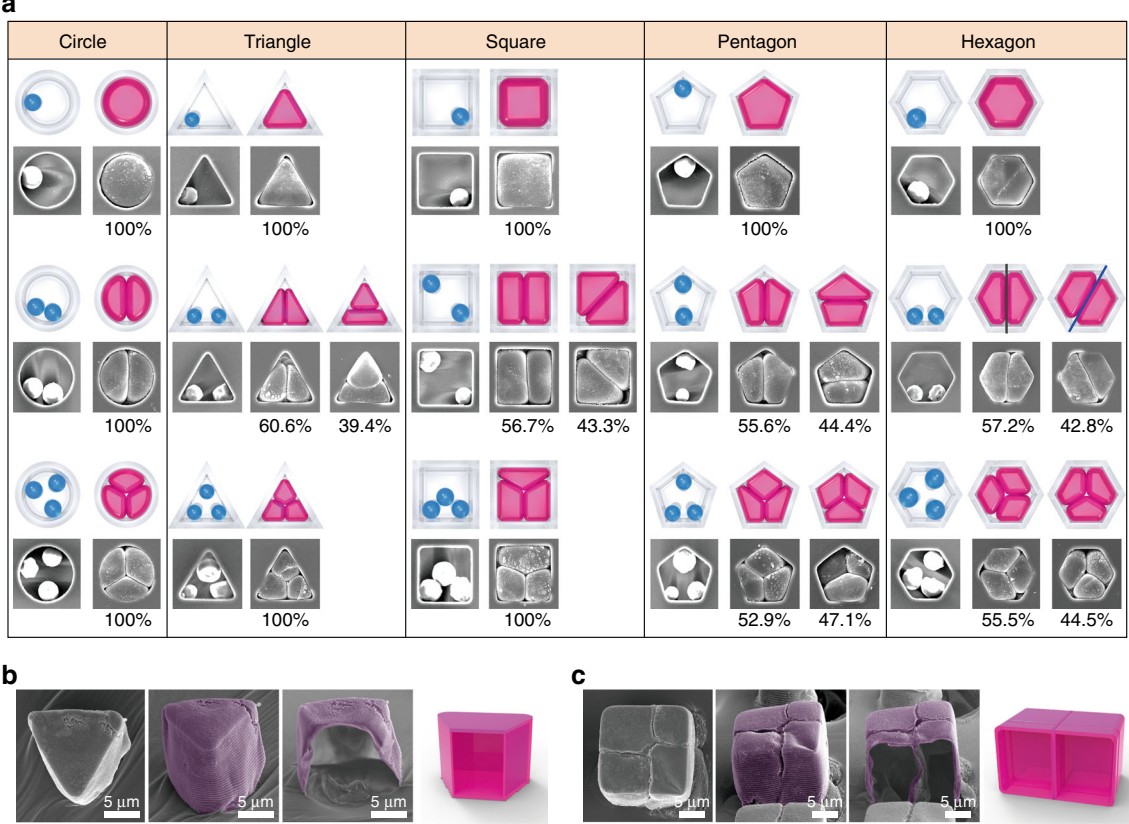

**Fig. 5** A library of topographic clusters. **a** SEM images of the clusters and the corresponding illustrations. The number of cleavages of each cell corresponds to the number of the initial seeds. **b**, **c** Close-up SEM images of the triangular prisms (**b**) and four-cell cubes (**c**). The last image of each group shows the cavity within the clusters, for which they were cut by the FIB

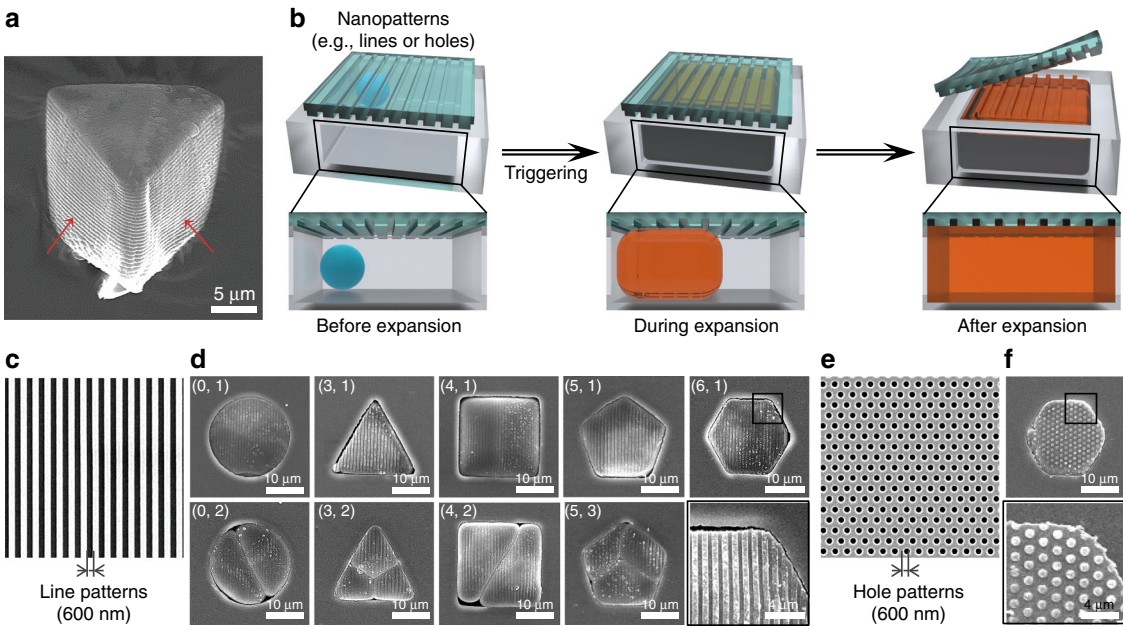

**Fig. 6** Hierarchical leveling on the topographic clusters. **a** SEM image of a single-cell microbomb with nanoscallops on the sidewall. **b** Nanoimprinting on the surface of the cluster particle to achieve further hierarchy. **c**, **d** SEM images of the nanoline mold (600 nm line width, **c**) and the clusters imprinted with nanolines on top (**d**). **e**, **f** SEM images of the mold of hole patterns (**e**) and the complementary hierarchical clusters with nano-pillars (600 nm in diameter and 600 nm in height) after the explosion (**f**)

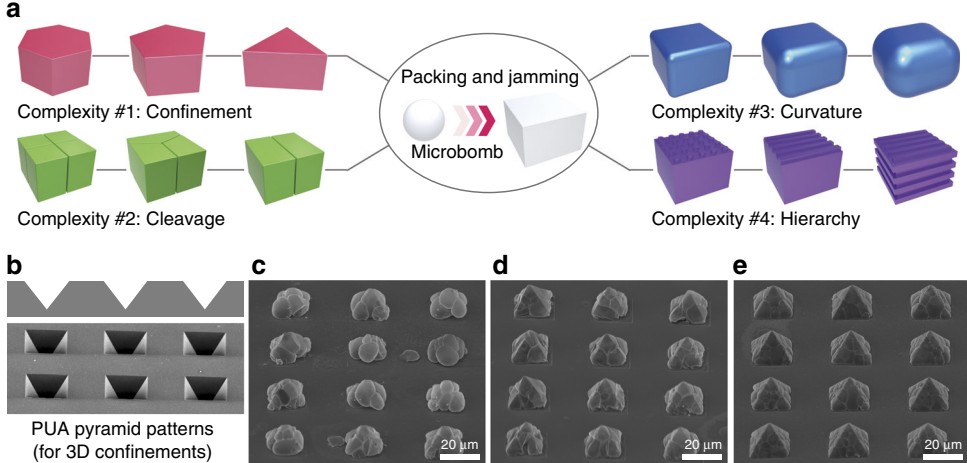

**Fig. 7** Topographical complexity of micro-clusters. **a** Illustration of the topographical complexity of the clusters as a result of the boundary shapes, the cleavages, the edge profiles, and the hierarchy. **b**–**e** Exemplary SEM images of the inverse pyramidal confinement and three-dimensional clusters formed within the micro-pyramids

filling the microwell, imitating the shape of the confinement. For $N > 1$, the differentiated cleavages were symmetric when $k = V/N$ yielded an integer $k$, i.e., $(V, N) = (0, 2), (0, 3), (3, 3), (4, 2), (6, 2)$, and $(6, 3)$. Otherwise, clusters with random microbomb assembly were observed. Regarding the orientation of the constituent cells, the $(6, 2)$ combination, for example, could program two types of symmetric clusters: one with a cleavage wall normal to one of the 6 edges where the interior angle of a unit is $3\pi$ (marked as a black line), and the other containing two isosceles trapezoids (internal angle is $2\pi$) shares the space diagonally (marked as a blue line). In addition, it is important to note that since there are no predictable cues for the cleavage orientation in the case of the singularity ($V = 0$), the perfect programming of clusters (100%) for all cases of $N$ is warranted from the non-pinning mobility in confinement. The SEM images in Fig. 5b, c reveal the hollow nature of the clusters from a triangular prism (Fig. 5b) and a cube with a cleavage connecting four cells (Fig. 5c) after focused ion beam (FIB) milling. At room temperature, the shell became glassy again, and thus the microbombs could maintain the packed shape permanently.

**Topographical imprinting of microbombs**. The close-up SEM image of the triangular prism reveals nanostructures on the side walls after the explosion as shown in Fig. 6a (red arrows), where such pattern was molded from the side wall of the PUA well. The PUA confinement well was replicated from the Si master with nanoscallops on the side wall as the result of the deep reactive-ion-etching process. During the explosion, the internal pressure exceeded 2500 kPa, high enough to push the viscoelastic shells toward the scallops on the side walls in confinement. With this in mind, we then introduced structural hierarchy to the clusters by covering the seeds with a PUA mold with well-defined nano-patterns as the top layer (Fig. 6b). Here all requirements for conventional nanoimprint lithography (NIL) could be satisfied by the thermomechanical properties of the microbombs, including viscoelasticity of the shell at $T > T_g$ and a sufficiently high internal pressure to enforce the mobility of the polymer chains toward the nano-patterns. As the microbomb features dynamic bulging and expansion, a physical collision occurs inside the confinement, in turn inscribing complementary nanostructures on the microbomb against the predefined patterns of the confinement. For example, we could engrave nanolines (600 nm for both width and spacing, Fig. 6c, d) onto the top surfaces of clusters, while nanopillars were created on the clusters by

imprinting against nanoholes (600 nm in diameter; Fig. 6e, f). Since the surface of the pristine microbomb is rough, imprinting on the clusters offers an alternative way to create smooth facets together with introducing hierarchical cleavages. Previously, we demonstrated the creation of structural hierarchy within flexible, free-standing membranes[37, 38] together with multilevel structures[33]. Here topographic assemblies with a hierarchy are obtained by nanoimprinting the microbombs during their expansion and topographic clustering.

## Discussion

To highlight the capability toward topographic clustering by inverse jamming and packing of microbombs, we illustrated a set of design rules on complexity in Fig. 7a, including the mold shapes ($V$) for confinement, the number of seeds ($N$) and partitioning in the cluster to design cleavages, the sharpness of edges around the cluster, and the hierarchy. We could adopt a library of molds fabricated by soft lithography for physical guidance during the microbomb explosions, which would be useful for exploring the diversity of forms in the final clusters with predesigned vertices $V$. Clusters with circular, triangular, square, pentagonal, and hexagonal cross-sections have been presented via the in situ, one-step explosion process in this paper. Furthermore, the complexity could be differentiated between clusters with the same vertices by modulating the number of initial seeds (that is, creating cleavages). Depending on the shape of the confinement well, symmetric or asymmetric partitioning of the clusters occurred, and the adjacent microbombs were integrated side by side with no interstitial voids as a result of the inverse jamming and topographically close-packing. Moreover, the edge profile could be modulated by the trigger temperature; sharp edges were obtained at higher temperatures (Supplementary Figs. 7 and 8). Finally, additional complexity and hierarchy could be introduced by imprinting the surfaces of the clusters. Although not a focus here, 3D shaping of the clusters should be possible. As seen in Fig. 7b–d, we prepared pyramidal clusters with various morphologies by precisely controlling the degree of expansion. Further, the hollow and ultralight weight nature of the clusters enables facile collection of them with batch-to-batch processability as shown in Supplementary Figs. 9 and 10, where $\sim 6.25 \times 10^6$ particles were obtained by the one-step process using the well size of $20\,\mu m \times 20\,\mu m \times 20\,\mu m$ with $1:1$ spacing ratio from the $10 \times 10\,cm^2$ patterned area. The adhesive interactions between the viscoelastic shells could be further modulated by

applying a low friction coating, allowing for dissembling the unit seeds from the assembled clusters (Supplementary Fig. 11). We believe that the microbombs offer a model system of adaptive building blocks to design more complex yet functional and ultra-lightweight particles by clustering via jamming and packing.

## Methods

**Microfluidic sorting of microbombs**. The microfluidic platform was fabricated by following the designs in literature[36] for sorting microbombs, which have broad size distribution as received. A PDMS microchannel was replicated from the Si master and connected to a glass slide after oxygen plasma treatment for 1 min (Femto Science Inc., Cute 09033C). After connecting tubes to both the inlet and the outlets, we introduced the microbomb solution at a flow rate of 1 mL min$^{-1}$ using a syringe pump (Pump 11 Elite, Harvard Apparatus, Holliston).

**Anchoring microbombs for topographic confinement**. A mixture of PDMS (Sylgard 184, Dow Corning, MI, USA) and curing agent (10:1 wt/wt) was poured onto a Si master patterned with an array of microwells (20 μm in width and 15 μm in height) prepared by photolithography, followed by deep reactive-ion-etching. The cured PDMS mold was placed onto the PUA311 (Minuta Tech, Gyeonggi-do, Republic of Korea) precursor resins, followed by flood ultraviolet curing at 365 nm. The microbombs (Expancel 461 DU40, Akzo Nobel, Amsterdam, Netherlands) were purchased and dispersed in water (1 mg mL$^{-1}$). A PUA channel (2 mm in width, 30 μm in height, and 30 mm in length; Supplementary Fig. 5) was placed over the patterned microwells, followed by introduction of the aq. solution of the microbombs via the channel. The outlets of the channels were first sealed with a tape to prevent water evaporation. After completion of the droplet filling, the water was allowed to evaporate, and the receding of the meniscus led to the microbombs anchoring within the microwells. Then, the guide PUA channel was peeled off from the PUA microwells.

**Inverse jamming and topographic packing of microbombs**. After the anchoring, the PUA wells were covered with a blanketing film (a flat PDMS film or a nano-patterned PUA film) to construct a closed confinement system against the explosion. We applied pressure (~ 1.5 MPa) on top of the blanket to prevent the microbombs escaping via the contact interface. To control the shape of the clusters, the sample was heated to different temperatures and held for 5 min to allow the microbombs to complete the explosion in the confinement. The resulting clusters were quenched by liquid nitrogen to fix their shape without mechanical shrinkage for observations. To release the clusters, water-soluble adhesive tape (5414, 3M, MN, USA) was used to remove the clusters from the micro-confinement. After dissolving the tape with water, the individual clusters floated on the water due to their ultralight weight properties and were collected for SEM imaging.

**Thermomechanical analysis and tracing the microbomb expansion**. The expansion characteristics of microbombs at different temperatures were measured using DSC (Q20, TA Instruments, DE, USA), TGA (Q50, TA Instruments), and optical microscopy (OM, LM-2500, Leica, Wetzlar, Germany). Specifically, the thermomechanical transition of the PVCAMM shell of the microbombs was characterized during heating from 30 to 150 °C using DSC. During the isothermal heating, the volume changes and the trajectory of the microbombs were monitored in situ using OM. The traces of the expansion were plotted using ImageJ (NIH). A FIB (Quanta 3D, FEI, OR, USA) combined with SEM was utilized to cut the clusters and observe the inside cavity using an accelerating voltage of 30 keV. For high-resolution images, field-emission SEM (Inspect F50, FEI) was utilized with an accelerating voltage of 10 keV.

**Data availability**. The authors declare that the data supporting the findings of this study are available within the paper and its Supplementary Information files.

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

## Acknowledgements

This work was supported by Fundamental R&D Program for Core Technology of Materials and the Industrial Strategic Technology Development Program funded by the Ministry of Trade, Industry and Energy, Republic of Korea. This work was also partially funded by Korea Institute of Science and Technology through Young Fellow program. This research was also supported by Basic Science Research Program through the National Research Foundation of Korea (NRF) funded by the Ministry of Science, ICT & Future Planning. This research was also supported by a grant from the Disaster and Safety Management Institute funded by the Ministry of Public Safety and Security of Korea government. S.Y. acknowledges support from National Science Foundation (NSF) Emerging Frontiers in Research and Innovation-Origami Design for Integration of Self-Assembling Systems for Engineering Innovation (NSF/EFRI-ODISSEI) Grant No. EFRI 13–31583.

## Author contributions

S.Y. and H.C. contributed equally to this work. C.M.K. and S.Y. led this work. S.Y., H.C., J.P.H., H.P., J.C.J., H.S.K., J.K., and S.H.L. performed experiments. S.Y., H.C., A.S.L., S.Y., and C.M.K. wrote the manuscript. S.M.H., J.H.L., and C.P. analyzed data.

## Additional information

**Competing interests:** The authors declare no competing financial interests.

