## [Peer Review File · Nature Communications]

Reviewer #1:

Remarks to the Author:

In this paper, the authors presented a novel approach to designing and fabricating micro-clusters with desirable geometrical characteristics, which can be subsequently employed as building blocks for self-assembly of larger-scale materials. Specifically, the authors used microbombs as seeds in confining microwells. The microbombs are composed of a viscoelastic polymeric shell and a volatile gas core, and possess a unique property of a thermal volumetric expansion up to 3000%. For a given number of microbomb seeds in a confining microwell, the authors further exploited the topological close-packing and kinetic jamming of the microbombs to fabricate microclusters of thermally expanded microbombs with the same shape of the microwells, which correspond to certain (symmetric) tiles of the microwell. A rigorous design rule including microwell shape, seed number, cluster edge sharpness and structural hierarchy has been put forth.

The paper is of great topical interest, technically sound and overall well written. The conclusions are well supported. It could be published in Nat Comm. However, the authors should consider addressing the following concerns and suggestions:

It seems to the reviewer that the key mechanism for fabricating the diversity of different shaped micro-clusters is the large thermal expansion of the microbombs. Based on this mechanism, it seems that the most efficient and straightforward procedure to fabricate different shaped particles is to use a single seed (i.e., $N = 1$). Maybe I've missed this point, but it was not clear what was the advantage of using multiple seeds?

In the "2D" case (i.e., when the micro-cluster is a prism), it is easier to control the symmetry the "jammed" packing of the seeds in the confine to generate the desirable micro-clusters. This seems to be much more difficult in 3D, as the 3D packing structures even composed of a small number of microspheres include a much larger number of asymmetric configurations that significantly overwhelms that few symmetric configurations (which are entropically unflavored). This again makes it more reasonable to use single seed instead of multiple seeds in 3D.

It might be helpful to (quantitatively) determine the interactions between the micro-clusters, especially those composed of multiple seeds. This would be extremely valuable for understanding their self-assembly behaviors.

Reviewer #2:

Remarks to the Author:

This manuscript presents a novel method for making particles with a variety of shapes. While there are many papers published daily that report new particle shapes, this particular method is unusual and extremely interesting. My reaction upon seeing the figures was, "Now THAT is cool!" From what I understand, a template is fabricated, spherical particles are introduced to the template well, the spheres are "exploded" and fuse together within the well to create a new particle with a nonspherical shape. By controlling the numbers of spheres, e.g., the authors can dial in the ultimate particle shape. Moreover, the resulting "cluster particles" are hollow. I found the paper super interesting and think it fits well into Nature Communications, but I have several questions for clarification, and the manuscript should be edited by a native English speaker for improved clarity and readability. My questions/comments are:

1. What is the range of sizes possible for the resulting particles? From the images, the particles seem big, too large to be Brownian and undergo self-assembly into structures comprised of many particles.

Is this correct?

2. I suppose it's ok to call the process by which the exploded spheres merge "jamming", but jamming usually refers to hard particles, not deformable particles. Do i understand correctly that the spheres, upon exploding, coalesce with the other particles within the well and thus deform? In the end, the images made me think more of a foam than granular media.

3. Is the fabrication process scalable? How many particles could practically be made using this approach? Is the process expensive?

4. I think the authors are using jamming both to describe how exploded spheres in a well make a new particle, but also what one could do with many such particles once made. Is that correct? What properties do they expect a jammed material comprised of these new particles to have?

REVIEWERS' COMMENTS:

Reviewer #1 (Remarks to the Author):

In this revised manuscript, the authors have constructively addressed the comments and suggestions from both reviewers as well as from the editors. The rationale for using multiple seeds for scale-up fabrication is well appreciated. The paper can be published in its current form.

Reviewer #2 (Remarks to the Author):

The authors have satisfactorily addressed my questions and I recommend the paper for publication in Nature Communications. With respect to the use of the terms "jamming" and "packing", the figure supplied in the response document, but not included in the revised manuscript or SI, nicely helps to explain their use of the terms in the context of two stages of cluster formation. It should be included in the final paper somewhere. I assume the transition from jamming to packing here occurs once the exploding particles touch both each other and the boundaries of the confining well, and can no longer expand without deformation. If so, then i would redraw the second image in the figure to reflect this, or add a third image to the figure. With regards to packing, typically packing refers to the packing of hard objects, not deformable ones, and when packing deformable objects using the entire expression "packing of deformable objects" is used. Perhaps the authors can be more precise here.

Response to the reviewers' comments

(Editor's comments are in black and authors' response in blue)

Reviewer #1 (Remarks to the Author):

In this paper, the authors presented a novel approach to designing and fabricating micro-clusters with desirable geometrical characteristics, which can be subsequently employed as building blocks for self-assembly of larger-scale materials. Specifically, the authors used microbombs as seeds in confining microwells. The microbombs are composed of a viscoelastic polymeric shell and a volatile gas core, and possess a unique property of a thermal volumetric expansion up to 3000%. For a given number of microbomb seeds in a confining microwell, the authors further exploited the topological close-packing and kinetic jamming of the microbombs to fabricate microclusters of thermally expanded microbombs with the same shape of the microwells, which correspond to certain (symmetric) tiles of the microwell. A rigorous design rule including microwell shape, seed number, cluster edge sharpness and structural hierarchy has been put forth.

The paper is of great topical interest, technically sound and overall well written. The conclusions are well supported. It could be published in Nat Comm. However, the authors should consider addressing the following concerns and suggestions:

■ We appreciate the reviewer's very positive comments and recommendation for publication of this paper. To address the reviewer's concerns, we thoroughly revised our manuscript according to the reviewer's comments.

Q1. It seems to the reviewer that the key mechanism for fabricating the diversity of different shaped micro-clusters is the large thermal expansion of the microbombs. Based on this mechanism, it seems that the most efficient and straightforward procedure to fabricate different shaped particles is to use a single seed (i.e., $N = 1$). Maybe I've missed this point, but it was not clear what was the advantage of using multiple seeds?

■ We agree that employing a single seed ($N = 1$) is the most efficient and straightforward way to exploit the topological invariant during the thermal expansion of microbomb, for shaping particles within the programmed confinement. Here, by using multiple seeds ($N > 1$), we offered an easy access to, and more importantly, a method to scale up the fabrication (see Figure S9) of differently shaped particles by the topological invariant as shown in Figure 5a.

■ It also allowed us to investigate the physical interactions of several microbombs in contact, which leads to rich variety of strategies to design and program shape complexity in hollow colloidal systems as summarized in Figure 7.

■ Further, the resulting colloidal particle systems could be mechanically more stable against the external forces much like to the honeycomb structure in nature.

Supplementary Figure S9 | Scalable fabrication of topographic clusters shaped in cubes by exploiting multiple seed expansion. (a, b) Microclusters floating on water after dissolving the water-soluble tape. Note that the hollow nature of our clusters leads to the facile collection of them as shown in the top-view image (b). (c-f) SEM images of collected clusters and their morphologies. Note that multiple seed anchoring and expansion enables scalable fabrication of $\sim 6.25 \times 10^6$ colloidal particles shaped in cubes after a single batch process. The well size is $20 \mu\text{m} \times 20 \mu\text{m} \times 20 \mu\text{m}$ with 1:1 spacing ratio from the $10 \times 10 \text{cm}^2$ patterned area.

■ The changes can be found in the revised manuscript (page 11 in the main text and page 12 in the supplementary information)

Q2. In the "2D" case (i.e., when the micro-cluster is a prism), it is easier to control the symmetry the "jammed" packing of the seeds in the confine to generate the desirable micro-clusters. This seems to be much more difficult in 3D, as the 3D packing structures even composed of a small number of microspheres include a much larger number of asymmetric configurations that significantly overwhelms that few symmetric configurations (which are entropically unflavored). This again makes it more reasonable to use single seed instead of multiple seeds in 3D.

It might be helpful to (quantitatively) determine the interactions between the micro-clusters, especially those composed of multiple seeds. This would be extremely valuable for understanding their self-assembly behaviors.

■ We agree with the reviewer in the case of the multiple seed confinement in 3D (e.g., big pyramid), expansion of each seed could result in entropically unfavorable configurations, leading to randomly oriented units in the cluster. In the meantime, the interaction between the seeds is crucial to accommodate their orientation. To address this, we investigated the inverse problem that is the possibility of

disassembly of the packed clusters, by reducing the adhesive interaction of each seed during expansion. We applied lubricant on the seed surface and monitored the expansion of multiple seeds in confinement by precise control of the temperature and time. Thereby we were able to disassemble the expanded units via mechanical detachment. Interestingly, it allowed us to revisit the reviewer's comment above (*i.e.*, $N = 1$) even we performed the experiment from the multiple seeds condition ($N > 1$). That is our strategy allowed us to obtain asymmetric colloidal particles even though symmetric confinement was used, see half-moon particles from the circular confinement as shown in Figure S10 (a) and (b), and half-pentagon particle in Figure S10 (c).

Supplementary Figure S10 | SEM images showing partially disassembled microclusters by reducing the surface-to-surface interaction. Lubricant was applied to the seed particles prior to expansion to prevent the bonding between seeds.

■ The changes can be found in the revised manuscript (page 11 in the main text and page 13 in the supplementary information)

“The adhesive interactions between the viscoelastic shells could be modulated by application of a low-friction coating, allowing for disassembling the unit seeds from the assembled clusters (Supplementary Fig. S10).”

Reviewer #2 (Remarks to the Author):

This manuscript presents a novel method for making particles with a variety of shapes. While there are many papers published daily that report new particle shapes, this particular method is unusual and extremely interesting. My reaction upon seeing the figures was, "Now THAT is cool!." From what I understand, a template is fabricated, spherical particles are introduced to the template well, the spheres are "exploded" and fuse together within the well to create a new particle with a nonspherical shape. By controlling the numbers of spheres, e.g., the authors can dial in the ultimate particle shape. Moreover, the resulting "cluster particles" are hollow. I found the paper super interesting and think it fits well into

Nature Communications, but i have several questions for clarification, and the manuscript should be edited by a native English speaker for improved clarity and readability.

- We appreciate the reviewer for mentioning our work as unusual and extremely interesting, as well as recommendation for publication of this paper.

- To address the reviewer's concerns, we revised our manuscript, and the final manuscript has been edited by the NPG Language Editing Service, to satisfy the concerns that the reviewer pointed out.

My questions/comments are:

Q1. What is the range of sizes possible for the resulting particles? From the images, the particles seem big, too large to be Brownian and undergo self-assembly into structures comprised of many particles. Is this correct?

- It is correct that the seeds used in this work are too big (6 ~ 15 μm in diameter) to be Brownian. To exploit our strategy for packing and jamming of microbombs, we used commercially available expandable polymer microcapsules as a model system for convenience. Here, we did not claim self-assembly of these particles via Brownian interaction. Of course, smaller Brownian particles can be synthesized to investigate the self-assembly process.

- Smaller sized bombs can be synthesized, for example, through several methods including suspension polymerization, emulsion polymerization and thermally induced phase separation methods. The most common method for synthesis of expandable microbead is suspension polymerization (for particles with a diameter of 1 μm to 1000 μm). However, emulsion polymerization and thermally induced phase separation could provide much smaller beads (for particles with a diameter of 70 nm to a few micrometers) as found in the literature [1-3].

[1] Guo, J. S., El-Aasser, M. S. & Vanderhoff, J. W. Microemulsion Polymerization of Styrene. *J. Polym. Sci., Part A: Polym. Chem.* **27**, 691-710 (1989).

[2] McDonald, C. J. *et al.* Emulsion Polymerization of Voided Particles by Encapsulation of a Nonsolvent. *Macromolecules* **33**, 1593-1605 (2000).

[3] Ogawa, H., Ito, A., Taki, K. & Ohshima, M. A New Technique for Foaming Submicron Size Poly(methyl methacrylate) Particles. *J. Appl. Polym. Sci.* **106**, 2825-2830 (2007).

Q2. I suppose it's ok to call the process by which the exploded spheres merge "jamming", but jamming usually refers to hard particles, not deformable particles. Do i understand correctly that the spheres, upon exploding, coalesce with the other particles within the well and thus deform? In the end, the images made me think more of a foam than granular media.

- We appreciate the reviewer's valuable comment that will strengthen our manuscript. In the original manuscript, we mixed two terminologies, packing and jamming, to simplify the plastic deformation of microbombs in the physical confinement. As the reviewer pointed out, jamming is typically for reference to hard particles, whereas our system is indeed the inverse process; the density increases as the microbombs expand in fixed boundaries. For the better understanding of expansion process of the seeds, we'd like to differentiate the inverse jamming and packing steps (see illustration below).

i) Inverse jamming (previously referred as kinetic jamming due to the kinetic and potential expansion of microbombs): at the very early expansion stage, analogous to the jamming inversely, the microbombs begin to expand, leading to contact between expanded microbombs.

ii) Topographic packing: at the second stage once microbombs touch the wall of the confinement, anisotropic expansion occurred, displaying the close packing along with the boundary deformation like foams in unyielding walls. Note that jamming might occur continuously, but packing is much serious at this stage, which can increase anisotropy in each unit.

Figure for reviewers only. Differentiation of inverse jamming and packing processes during the thermal expansion of the microbombs.

■ In the revision, we revised the title of our manuscript as

“Shaping micro-clusters via inverse jamming and topographic close-packing of microbombs” to prevent the confusion.

■ We further delineate the jamming and packing stage in the revised manuscript (page 8)

“Within each well, while the total volume fraction ϕ of the cluster increased with the expansion of the individual seeds (*i.e.*, inverse jamming)”

“Hence, the paired seeds could show limited mobility yet still be active to the direction that has more free space. Soon, severe stress is applied to the surface in contact, topographically deforming the shapes of seeds to yield dense packing similar to the foams in expansion.”

“When the two seeds were in contact initially (paired), they expanded isotopically in 2D (discotic) as they are free even the contact maintained ($t \sim 10$ s, Fig. 4c). The cluster then asymmetrically expanded side-by-side by deforming the shells to fulfill the dense-packing in the well, forming two rectangular cells ($t \sim 92$ s). The evidence of jamming is much clear in addressing initial interaction between the two separated seeds as seen in Fig. 4d (see the black and red circles, < 5 s).”

Q3. Is the fabrication process scalable? How many particles could practically be made using this approach? Is the process expensive?

■ One of the biggest advantages in utilizing the commercially available expandable polymer beads is that they are scalable. See response to Reviewer 1, we added Supplementary Figure S9 to demonstrate the scalability, forming $\sim 6.25 \times 10^6$ cubic clusters in a single batch, using wells sized as $20 \mu\text{m} \times 20 \mu\text{m} \times 20 \mu\text{m}$ with 1:1 spacing ratio from the $10 \times 10 \text{cm}^2$ patterned area.

Supplementary Figure S9 | Scalable fabrication of topographic clusters shaped in cubes by exploiting multiple seed expansion. (a, b) Microclusters floating on water after dissolving the water-soluble tape. Note that the hollow nature of our clusters leads to the facile collection of them as shown in the top-view image (b). (c-f) SEM images of collected clusters and their morphologies. Note that multiple seed anchoring and expansion enables scalable fabrication of $\sim 6.25 \times 10^6$ colloidal particles shaped in cubes after a single batch process. The well size is $20 \mu\text{m} \times 20 \mu\text{m} \times 20 \mu\text{m}$ with 1:1 spacing ratio from the $10 \times 10 \text{cm}^2$ patterned area.

■ The approximate cost for the single batch process we performed is less than \$1.00 which is competitive with other process for synthetic particles.

Q4. I think the authors are using jamming both to describe how exploded spheres in a well make a new particle, but also what one could do with many such particles once made. Is that correct? What properties do they expect a jammed material comprised of these new particles to have?

- Yes, it is correct that we make new particles by expanding the spheres in a well. These hollow particles are extremely light-weight yet, could be mechanically strong much like the honeycomb structure in nature.
- As we responded to reviewer 1, here, we investigated one more interesting strategy to exploit the interaction during jamming and packing, in turn enabling revisiting the case of $N=1$ after disassembling the clusters. To address this, we utilized lubricant to decrease the surface interaction of jammed clusters during packing, which leads to the disintegration of unit seeds after the process.

Supplementary Figure S10 | SEM images showing partially disassembled microclusters by reducing the surface-to-surface interaction. Lubricant was applied to the microcapsules prior to expansion to prevent the bonding between seeds.

- The significance of this strategy is that we could obtain asymmetric colloidal particles even though symmetric confinement was selected (i.e., half-moon particles from the circular confinement as shown in (a) and (b)), and half-pentagon particle in Figure S10 (c)).
- The changes can be found in the revised manuscript (page 11 in the main text and page 13 in the supplementary information)

“The adhesive interactions between the viscoelastic shells could be modulated by application of a low-friction coating, allowing for disassembling the unit seeds from the assembled clusters (Supplementary Fig. S10).”

- Beside the various shapes of clusters, hollow characteristic is one of the unusual advantages of our strategy, enabling facile collection of them on top surface of water after batch processes as shown in the Supplementary Figure S9.

Point-by-point response to the reviewers' comments

Reviewer #1

In this revised manuscript, the authors have constructively addressed the comments and suggestions from both reviewers as well as from the editors. The rationale for using multiple seeds for scale-up fabrication is well appreciated. The paper can be published in its current form.

= > We greatly appreciate the reviewer's decision.

Reviewer #2

The authors have satisfactorily addressed my questions and I recommend the paper for publication in Nature Communications. With respect to the use of the terms "jamming" and "packing", the figure supplied in the response document, but not included in the revised manuscript or SI, nicely helps to explain their use of the terms in the context of two stages of cluster formation. It should be included in the final paper somewhere. I assume the transition from jamming to packing here occurs once the exploding particles touch both each other and the boundaries of the confining well, and can no longer expand without deformation. If so, then i would redraw the second image in the figure to reflect this, or add a third image to the figure. With regards to packing, typically packing refers to the packing of hard objects, not deformable ones, and when packing deformable objects using the entire expression "packing of deformable objects" is used. Perhaps the authors can be more precise here.

= > We greatly appreciate the reviewer's suggestion. Based on the comment the reviewer mentioned, we revised the main manuscript and included the modified illustration as new Supplementary Figure 6 in the revised manuscript.